# Prolonged COVID-19 Pneumonia in Patients with Hematologic Malignancies: Clinical Significance and Serial CT Findings

**DOI:** 10.3390/jcm14082701

**Published:** 2025-04-15

**Authors:** Dae Hee Han, Raeseok Lee, Gi June Min, Jongmin Lee, Yejin Sohn, Eun Jeong Min, Jinyoung Lee, Jung Im Jung, Kyongmin Sarah Beck

**Affiliations:** 1Department of Radiology, Seoul St. Mary’s Hospital, College of Medicine, The Catholic University of Korea, 222 Banpo-daero, Seocho-gu, Seoul 06591, Republic of Korea; lepolder@gmail.com (D.H.H.); yngl@kakao.com (J.L.); jijung@catholic.ac.kr (J.I.J.); 2Division of Infectious Diseases, Department of Internal Medicine, Seoul St. Mary’s Hospital, College of Medicine, The Catholic University of Korea, 222 Banpo-daero, Seocho-gu, Seoul 06591, Republic of Korea; misozium03@gmail.com; 3Department of Hematology, Seoul St. Mary’s Hospital, College of Medicine, The Catholic University of Korea, 222 Banpo-daero, Seocho-gu, Seoul 06591, Republic of Korea; beichest@nate.com; 4Division of Pulmonary and Critical Care Medicine, Department of Internal Medicine, Seoul St. Mary’s Hospital, College of Medicine, The Catholic University of Korea, 222 Banpo-daero, Seocho-gu, Seoul 06591, Republic of Korea; dibs03@gmail.com; 5Department of Medical Sciences, The Catholic University of Korea, Graduate School, 222 Banpo-daero, Seocho-gu, Seoul 06591, Republic of Korea; sohn.yejin.9@gmail.com; 6Department of Medical Life Sciences, College of Medicine, The Catholic University of Korea, 222 Banpo-daero, Seocho-gu, Seoul 06591, Republic of Korea; ej.min@catholic.ac.kr

**Keywords:** COVID-19, hematologic malignancy, prolonged SARS-CoV-2 infection, chest CT, lymphoma, rituximab, teclistamab

## Abstract

**(1) Background:** Hematologic malignancy patients have a heightened risk for prolonged COVID-19 pneumonia. **(2) Methods:** We retrospectively investigated the clinical significance and serial CT findings of prolonged COVID-19 pneumonia in hematologic malignancy patients. Hematologic malignancy patients with persistent SARS-CoV-2 polymerase chain reaction (PCR) positivity >30 days and more than one chest CT after initial positivity were reviewed. Serial CT images were analyzed for the presence of COVID-19 pneumonia, patterns and distribution of CT findings, and severity scores of lung involvement. Clinical characteristics of the patients, including treatments for underlying hematologic malignancy prior to and after COVID-19 and COVID-19-related factors, were compared according to the presence of COVID-19 pneumonia. **(3) Results:** A total of 55 patients (36 male, median age 60 years) were included in the study. A total of 56.4% had received B-cell-directed therapies, such as rituximab or teclistamab, within one year of COVID-19 diagnosis. A total of 76.4% of patients had the presence of COVID-19 pneumonia on CT, with a median CT duration of pneumonia of 35.5 days, and they experienced more frequent (*p* = 0.005) and longer (*p* = 0.002) hospital stays and longer delays in treatment for underlying malignancy (*p* = 0.03), compared to those without evidence of COVID-19 pneumonia on CT. The development of COVID-19 pneumonia was significantly related to B-cell-directed antibody therapies (*p* = 0.02). Median CT severity scores during <30 days, 30–59 days, 60–89 days, and ≥90 days from initial diagnosis were 2.0, 2.0, 2.0, and 1.0, respectively. **(4) Conclusions:** Patients with hematologic malignancies may experience prolonged COVID-19 pneumonia, which is associated with the use of B-cell-directed antibody-based drugs and can result in longer hospital stays and delays in treatments for underlying malignancy.

## 1. Introduction

Four years have passed since the initial wave of the COVID-19 pandemic, but the virus continues to plague many individuals worldwide, now as an endemic with periodic surges. Beyond the typical cases of COVID-19, there are cases in which the infection persists for over 30 days, known as persistent SARS-CoV-2 infection, characterized by ongoing virus replication and particularly affecting immunocompromised patients [1]. Patients with hematologic malignancies are at a heightened risk of developing this unique form of SARS-CoV-2 infection with unusually long duration, which may result from the complex interplay between the malignancy itself and multiple anticancer drugs and procedures that profoundly alter the immune system [1,2,3,4]. Many of the anticancer drugs that target the immune system and are used for the treatment of hematologic malignancies are known to be related to impaired immune responses after vaccination against SARS-CoV-2 [5,6,7,8], and one study found that patients with persistent SARS-CoV-2 infection had failed to develop detectable antibodies against SARS-CoV-2, despite vaccination [4]. Due to the impaired immune responses to SARS-CoV-2 in hematologic malignancy patients, the course of COVID-19 pneumonia also differs from that of immunocompetent patients, which typically regresses within two weeks [9,10,11]. There are many reports of immunocompromised patients showing prolonged COVID-19 pneumonia that could persist for as long as three months, especially in patients with impaired humoral (B-cell) immunity [4,12]. One systematic review of 24 immunocompromised patients with persistent SARS-CoV-2 infection suggested migrating pneumonia of intermediate severity that can persist for up to three months [4] to be the key CT findings, but the scope of this systematic review was limited, including patients both with and without hematologic malignancies, lacking comprehensive clinical details, and reviewing only a small number of representative published CT images [4]. Also, case reports and series may be prone to selection bias toward patients with nontypical imaging findings. Accurate knowledge of imaging findings of prolonged COVID-19 pneumonia in hematologic malignancies is important because CT detection of prolonged COVID-19 pneumonia in that context may lead to delays in or alterations of cancer treatment plans in these patients. Prolonged COVID-19 pneumonia may also increase the disease burden of hematologic malignancy patients, with its long treatments and potential interruptions in anticancer therapy. However, the clinical implications of prolonged COVID-19 pneumonia in this patient population remain inadequately explored. To address these gaps, a dedicated study focusing on the radiologic and clinical aspects of prolonged COVID-19 pneumonia in patients with hematologic malignancies is essential. We aimed to investigate the clinical significance and serial CT characteristics of prolonged COVID-19 pneumonia in hematologic malignancy patients.

## 2. Materials and Methods

This retrospective study was approved by the Institutional Review Board of Seoul St. Mary’s Hospital (approval number: KC23RISI0347). The requirement for written informed consent was waived.

### 2.1. Patients

The picture archiving and communication system at our institution was searched with the keywords “COVID” in English or “corona” in Korean from January 2020 to March 2023, and electronic medical charts were reviewed to select hematologic malignancy patients with COVID-19 and at least one chest CT after acquiring COVID-19. Of these patients, those with persistent SARS-CoV-2 positivity, which was defined as positive polymerase chain reaction (PCR) results for SARS-CoV-2 lasting more than 30 days, and more than two CT exams during the period of SARS-CoV-2 positivity, were included in this study. Exclusion criteria included patients without evidence of persistent SARS-CoV-2 positivity (either due to undergoing only one SARS-CoV-2 PCR study or SARS-CoV-2 positivity lasting less than or equal to 30 days); patients who were clinically suspected of reinfection rather than persistent infection, defined as new-onset symptoms with second PCR positivity after more than 45 days from first PCR positivity; and patients with only one chest CT taken during the period of SARS-CoV-2 positivity (Figure 1). Seven of the fifty-five patients had been previously reported in a case series [13], which was on migratory pneumonia with prolonged SARS-CoV-2 infection in lymphoma patients who had received anti-CD20 therapies. The current study expands on this by having a larger patient number that includes all patients with hematologic malignancy and persistent SARS-CoV-2 PCR positivity and analyzing the development of prolonged COVID-19 pneumonia.

### 2.2. Image Analysis

Two thoracic radiologists (with 8 and 23 years of experience, respectively) reviewed the serial chest CTs of the patients. First, each CT was categorized according to the Radiological Society of North America (RSNA) Expert Consensus Document (typical appearance, indeterminate appearance, atypical appearance, or negative for COVID-19 pneumonia) [14]. Then, for those with positive findings for COVID-19 pneumonia, the following items were assessed: CT density of pneumonia (ground-glass opacity [GGO], consolidation, or both); the predominant distribution of pneumonia (peripheral, peribronchovascular, or both); severity of pneumonia on a scale of 0–4 for the whole lungs (0, no involvement; 1, 1–25% involvement; 2, 26–50% involvement; 3, 51–75% involvement; and 4, 76–100% involvement); the interval changes in the overall extent of pneumonia (decreased, stable, or increased); and the presence of migration between two serial CTs, which was defined as the emergence of new airspace opacities in different lung regions accompanied by the resolution of previous lesions seen on earlier CT. A 0–4 scale CT severity scoring system was used because a previous systematic review on prolonged SARS-CoV-2 infection [4] utilized the same scoring system and found that severity scores remained persistently intermediate throughout the disease course. Given these findings, we anticipated similar results in our study and, therefore, did not consider a more detailed but complex scoring system necessary. A conclusion was reached for each case by consensus between the two radiologists. The total duration of pneumonia was calculated by counting the days between a patient’s first and last CTs with evidence of COVID-19 pneumonia.

### 2.3. Clinical Information Collection

Electronic medical charts of the study patients were reviewed. Factors related to underlying hematologic malignancy and factors related to COVID-19 were collected. The following factors related to underlying hematologic malignancy were reviewed by a hematologist: type of hematologic malignancy; last date and type of chemotherapy drugs; date of hematopoietic stem cell transplant; delays in treatment for hematologic malignancy; and estimated time of treatment delay in days. For chemotherapy drugs, the use of antibody-based drugs targeting the B-cell lineage was separately recorded, due to their reported treatment-related poor antibody responses after COVID-19 vaccination, their relationship between the treatment and persistent SARS-CoV-2 infection, increased infection risk [4,5,6,7,8,15], and our own clinical experience. Antibody-based drugs targeting the B-cell lineage included anti-CD20 agents (e.g., rituximab), anti-B-cell maturation antigen (BCMA) agents (e.g., teclistamab), an anti-CD38 agent (daratamumab), and an anti-CD22 antibody-drug conjugate (e.g., inotuzumab ozogamicin) [16]. The estimated time of delay was defined as the interval between scheduled date of treatment written in the medical charts and the actual application date. When the planned treatment could not be applied due to patient death, it was also considered a treatment delay, and the estimated time of delay was defined as the interval between the scheduled date and the date of death. The following factors related to COVID-19 were reviewed by an infectious diseases specialist: COVID-19 vaccination status; symptoms after 30 days from initial PCR positivity; number and dates of COVID-19 PCR; cycle threshold (Ct) values; presumed SARS-CoV-2 subtypes, based on the most dominant variant in South Korea at the date of initial PCR positivity [17,18]; number and duration of hospital admissions due to COVID-19; the use of antiviral and steroid therapies; the use of invasive ventilation; COVID-19-specific mortality; and 30-day all-cause mortality, which was defined as any death within 30 days from the last day of COVID-19 PCR positivity.

### 2.4. Statistical Analysis

Clinical and radiological factors were compared between those showing COVID-19 pneumonia (COVID-19 pneumonia group) and those without any evidence of COVID-19 pneumonia (no COVID-19 pneumonia group) during persistent SARS-CoV-2 positivity. The Mann–Whitney U test was used to analyze continuous variables, and Fisher’s exact test and the chi-square test were used for categorical variables, as appropriate. Univariable logistic regression analyses were performed to determine factors associated with the development of COVID-19 pneumonia. For multivariable logistic regression analysis, three variables with a *p* value of 0.10 or less were entered into the final model due to the small sample size. All statistical analyses were performed using SAS version 9.4 (SAS Institute, Cary, NC, USA).

## 3. Results

Of 286 patients with COVID-19 and hematologic malignancies at our hospital who underwent chest CT after acquiring COVID-19, there were 99 patients (34.6%) with persistent SARS-CoV-2 positivity. Among these patients, serial CTs during persistent SARS-CoV-2 positivity were available for review in 55 patients (36 male, median age 61 years).

### 3.1. Clinical Characteristics of Study Patients

The clinical characteristics of 55 included patients are summarized in Table 1. Lymphoma was the most common hematologic malignancy (54.5% [30/55]). In total, 65.5% (36/55) had received treatment with antibody-based agents targeting the B-cell lineage within one year of the diagnosis of COVID-19 (Table 2), of which 28 (50.9%) were anti-CD20 agents. The median duration of SARS-CoV-2 PCR positivity was 69 days (interquartile range [IQR] 55–100.5), the median number of PCR tests performed was 9 (IQR 5–11.5), and the median time interval between PCR tests was 7 days (IQR 6–13). A total of 96.3% (53/55) demonstrated median Ct value less than or equal to 30. Approximately 89.1% (49/55) complained of persistent or relapsing symptoms—including persistent fever, cough, sputum, dyspnea, sore throat, or myalgia—beyond 30 days from the first SARS-CoV-2 PCR positivity. Patients were treated with systemic corticosteroids (65.5% [36/55]) and antivirals such as remdesivir, nirmatrelvir/ritonavir, or molnupiravir (69.1% [38/55]). A total of 43.6% (24/55) of patients received a combination of systemic corticosteroids and antivirals. However, the dose, sequence, and combination of steroid and antiviral treatments varied greatly among study patients. COVID-19-specific mortality and 30-day all-cause mortality were 25.5% (14/55) and 38.2% (21/55), respectively.

### 3.2. Development of Prolonged COVID-19 Pneumonia

Patients underwent a median of three (IQR 2–4) CT scans during persistent SARS-CoV-2 positivity. A total of 76.4% (42/55) of study patients had evidence of COVID-19 pneumonia on any of the CTs during persistent SARS-CoV-2 positivity, and these patients were grouped as the “COVID-19 pneumonia group”. COVID-19 pneumonia was prolonged, with a median CT duration of pneumonia being 35.5 days (IQR 24.3–66.8). The median time of the last follow-up CT showing evidence of COVID-19 pneumonia from the initial COVID-19 diagnosis was 58.5 days (IQR 38.3–84.8). The remaining 23.6% (13/55) did not demonstrate any evidence of COVID-19 pneumonia on any of the CTs and were grouped as the “No COVID-19 pneumonia group”. Among these 13 patients, the lungs were clear in seven, while the other six showed CT findings consistent with fungal pneumonia. Among the six patients with fungal pneumonia, one had pathology-proven mucormycosis, and five showed multiple cavitary or noncavitary nodular opacities on chest CT with positive serum galactomannan results. Types of hematologic malignancies were significantly different between the COVID-19 pneumonia and the No COVID-19 pneumonia groups (*p* = 0.011); leukemia and multiple myeloma were a minority in the COVID-19 pneumonia and the No COVID-19 pneumonia groups, respectively. Significantly more patients in the COVID-19 pneumonia group had received B-cell-directed antibody-based agents within one year of COVID-19 diagnosis (*p* = 0.042). More patients with prolonged COVID-19 pneumonia were treated with systemic corticosteroids (*p* < 0.001). Prolonged COVID-19 pneumonia was associated with more frequent hospital admissions (*p* = 0.005), longer hospital stays for COVID-19 (20 days vs. 6 days, *p* = 0.002), and longer delays in treatments for hematologic malignancies (49 days vs. 29.5 days, *p* = 0.03) (Table 1). COVID-19-specific mortality (*p* = 0.147) and 30-day all-cause mortality (*p* > 0.999) did not show significant differences between the two groups. In the univariable logistic regression analysis (Table 3), only the use of antibody-based drugs targeting B-cell lineage within one year of COVID-19 diagnosis was significantly related to the development of COVID-19 pneumonia (*p* = 0.025, odds ratio [OR] = 4.51 [95% confidence interval: 1.21, 16.75]). Multivariable logistic regression analysis (Table 3) also revealed that only the use of antibody-based drugs targeting the B-cell lineage within one year of COVID-19 diagnosis was significantly associated with the development of COVID-19 pneumonia (*p* = 0.041, OR = 4.34 [95% confidence interval: 1.06, 17.81]).

### 3.3. Baseline and Follow-Up Chest CT Findings

The chest CT findings of COVID-19 pneumonia (N = 42), categorized into four groups (<30 days, 30–59 days, 60–89 days, ≥90 days from initial COVID-19 diagnosis) based on the period after initial diagnosis, are summarized in Table 4. The most common findings across all time intervals were a typical appearance of COVID-19 pneumonia, as defined by the RSNA CT categorization, and GGO with or without consolidation in both peripheral and peribronchovascular distribution. Median severity scores for CTs taken during <30 days, 30–59 days, 60–89 days, and ≥90 days from initial diagnosis were 2.0 (IQR 1.0–3.0), 2.0 (IQR 1.0–3.0), 2.0 (IQR 1.0–3.5), and 1.0 (IQR 1.0–2.0), respectively (Figure 2). Migration during prolonged COVID-19 pneumonia was observed in 20 patients (47.6%) (Figure 3 and Figure 4); these patients had used rituximab (N = 14), obinutuzumab (N = 1), odronextamab (N = 2), teclistamab (N = 2), elranatamab (N = 1), or venetoclax (N = 1) within one year of COVID-19 diagnosis. Across all time periods, variable changes in the extent of COVID-19 pneumonia (increased, stable, or decreased) were observed. The course of the radiologic opacities and the treatments that the patients received appeared to be unrelated, although a statistical analysis could not be performed to ascertain a causal relationship between treatments and airspace opacities due to the variability of the treatment regimens among the study patients.

## 4. Discussion

Our study comprehensively reviews the radiological and clinical profiles of prolonged COVID-19 pneumonia in patients with hematologic malignancies. Previous studies on the radiological findings of persistent SARS-CoV-2 infection and prolonged COVID-19 pneumonia have mostly been case reports and series, and this is the first study to analyze radiological findings in conjunction with clinical factors in a larger patient cohort.

Currently, there is no set definition of persistent SARS-CoV-2 infection, and different studies have used various definitions, including a specific duration of PCR positivity (21–56 days), presence of persistent or relapsing symptoms, Ct values, and radiological findings [1,2,4,19]. One study has proposed a Ct value of 30 or lower for 30 days or longer as a definition of persistent SARS-CoV-2 infection, as Ct values less than 30 are generally considered to be associated with isolation of infectious SARS-CoV-2 [1]. Except for two patients, all patients in our study demonstrated a median Ct value of less than 30 during prolonged SAR-CoV-2 positivity. However, among patients with a median Ct value of 30 or less, about 10% had no relapsing or persistent symptoms, and about 20% showed no evidence of COVID-19 pneumonia on CT. These findings highlight the difficulty of defining persistent SARS-CoV-2 infection.

Our study has demonstrated that not all patients with persistent SARS-CoV-2 positivity, including symptomatic patients, develop COVID-19 pneumonia. In fact, six of the 13 patients without COVID-19 pneumonia were diagnosed with fungal pneumonia. Given that a previous study reported a secondary diagnosis in 17% of COVID-19 patients [20], maintaining a high index of suspicion for coexisting conditions is crucial, as it is for prolonged SARS-CoV-2 infection. The development of COVID-19 pneumonia in this group of patients was significantly related to the use of antibody-based drugs targeting the B-cell lineage, which included not only rituximab and other anti-CD20 agents but also anti-BCMA antibodies such as teclistamab and elranatamab. Notably, COVID-19 pneumonia in this group of hematologic malignancy patients with persistent SARS-CoV-2 positivity was prolonged—typically lasting 1 to 3 months—in contrast to its regression within two weeks in immunocompetent patients with COVID-19 [9,10,11].

There are many reports of prolonged COVID-19 pneumonia, often in a unique migratory form, in patients who had received anti-CD20 agents [4,12], but this is the first study demonstrating prolonged COVID-19 pneumonia in patients treated with anti-BCMA agents. There have been reports of increased infection risk associated with the use of anti-BCMA bispecific antibodies for multiple myeloma [7,15], and one study has shown that teclistamab impairs humoral immunity, as demonstrated by the recipients’ inability to produce antibody responses against SARS-CoV-2 after vaccination [8]. Both BCMA and CD20 play critical roles in B-cell proliferation, activation, and survival [8,21], and persistent viral shedding into the lungs due to impaired humoral immunity caused by anti-CD20 and anti-BCMA agents may be responsible for prolonged COVID-19 pneumonia.

Radiologically, prolonged COVID-19 pneumonia was observed as mostly typical COVID-19 pneumonia in peripheral and peribronchovascular distribution, consisting of GGOs or mixed GGO and consolidation. These results are similar to—but slightly different from—those of the aforementioned systematic review of prolonged SARS-CoV-2 infection, which demonstrated that pneumonia in prolonged SARS-CoV-2 infection was seen as typical COVID-19 pneumonia with GGO in peripheral and peribronchovascular distribution [4]. Compared to that review, our study showed more cases of: (1) mixed GGO and consolidation (vs. mostly GGO), (2) less typical COVID-19 pneumonia patterns, and (3) more patients with severity scores of 3 or 4. These differences may be due to differences in study design; the systematic review evaluated only representative published images of the cases, whereas our study involved a review of the entire CT image sets of the patients. It could also be related to the longer follow-up period in our study.

Our study has documented the CT findings of prolonged COVID-19 pneumonia at 30–59 days, 60–89 days, and after 90 days from the diagnosis of COVID-19. The CT severity scores were persistently intermediate, and worsening of airspace opacities on CTs could be observed throughout the whole duration of the SARS-CoV-2 infection. Previous studies on the natural history of COVID-19 pneumonia in immunocompetent patients focused primarily on the first 30 days after diagnosis, with the peak known to occur around days 9–13 [9,10,11]. However, prolonged COVID-19 pneumonia appears to follow a very different course, and worsening (i.e., an increase in the extent of pneumonia on CT) occurs regardless of the time elapsed since diagnosis. Radiologists unfamiliar with prolonged COVID-19 pneumonia and its unique imaging progression may misinterpret CT aggravation several weeks after the initial diagnosis as a different condition. In patients with hematologic malignancies and a recent history of COVID-19 who show persistent airspace opacities on CT, particularly those who had received B-cell-directed antibody therapies, prolonged COVID-19 pneumonia should be considered as a possible diagnosis.

About half of the patients with prolonged COVID-19 pneumonia in our study demonstrated migration of airspace opacities, 80% of whom had received anti-CD 20 agents. The remaining four patients with migrating pneumonia had received either anti-BCMA agents or venetoclax, a B-cell lymphoma-2 (Bcl-2) inhibitor, all of which have been associated with diminished immune responses to SARS-CoV-2 vaccines [5,6,8], suggesting impaired immune responses to the virus. Migration of airspace opacities has been reported in patients with prolonged SARS-CoV-2 infection and impaired B-cell immunity, either due to B-cell depletion therapies or congenital immunodeficiency [4]. Some may argue that these migrating airspace opacities represent organizing pneumonia, as such patterns are also seen in that condition [22]. However, “reverse halo sign or other findings of organizing pneumonia” is included in the definition of the typical appearance of COVID-19 pneumonia, according to an expert panel review [14], and around 40% of autopsied patients with COVID-19 pneumonia demonstrate histopathologic features of organizing pneumonia. [23,24] This suggests that part of what we collectively describe as “COVID-19 pneumonia” may, in fact, be “SARS-CoV-2-induced secondary organizing pneumonia”. It is also important to note that “organizing pneumonia” is a common endpoint of lung injury caused by various insults [25]. The migrating airspace opacities seen in our study may reflect “organizing pneumonia” triggered by continuous viral shedding into the lungs; they should not be confused with cryptogenic organizing pneumonia or drug-induced organizing pneumonia. Migrating airspace opacities may be a characteristic imaging feature of prolonged COVID-19 pneumonia, potentially signifying ongoing viral shedding due to impaired B-cell-mediated immunity, although the exact mechanism remains unclear.

This study represents the first attempt to quantify the disease burden of prolonged COVID-19 pneumonia in hematologic malignancies. Although not all patients with persistent SARS-CoV-2 positivity developed prolonged COVID-19 pneumonia, those who did experience significantly longer stays and extended delays in the treatment of their underlying hematologic malignancies. At our institution, the decision to resume treatment for hematologic malignancies was made by infectious disease specialists. These decisions were based on a comprehensive evaluation of multiple factors, including SARS-CoV-2 PCR results, especially Ct values, clinical and radiologic improvement, and the intensity and urgency of the planned cancer therapy. When CT scans demonstrated the presence of prolonged COVID-19 pneumonia, further delays in cancer treatment were often warranted. There were also many cases in which delays in hematologic malignancy treatments led to the progression of the underlying diseases. Prolonged COVID-19 pneumonia exacerbates the already considerable disease burden of patients with hematologic malignancies.

Our study has several limitations. First, this was a single-center retrospective study with a relatively small sample size. However, prolonged COVID-19 pneumonia represents a clinically significant yet relatively rare condition, primarily affecting a highly selective group of immunocompromised patients. As a result, most available literature consists of case reports and the inclusion of 55 patients in our study represents a substantial cohort given the rarity of this condition. Our study provides valuable initial data on this underexplored topic, and future multicenter studies with larger cohorts are needed to validate our findings. Second, this study only included a subset of immunocompromised patients, excluding others such as solid organ transplant recipients. However, as many other studies focused solely on COVID-19 in hematologic malignancies suggest, this group represents a unique high-risk population due to prolonged immunodeficiency caused by both the malignancy and its treatments. Additionally, we excluded 44 patients with prolonged SARS-CoV-2 PCR positivity who had only one CT scan, which may have introduced selection bias. However, since our study aimed to analyze serial CT changes, the inclusion of patients with only a single scan was not feasible. Third, about half of the patients lacked information regarding COVID-19 vaccination status. As many of these patients are known to have impaired antibody responses despite vaccination [4,5], vaccination status likely did not make a significant difference in our cohort. Fourth, we recorded only the date of the last positive PCR result, and the information about the date of negative conversion of COVID-19 PCR could not be attained, which may have led to an underestimation of the duration of prolonged SARS-CoV-2 infection. However, as PCR testing was typically performed in symptomatic patients or those with persistent radiological abnormalities, the calculated duration of SARS-CoV-2 infection is assumed to be relatively accurate. Fifth, we were unable to perform statistical analyses on the relationship between changes in radiologic opacities and treatments. The substantial variability in treatment timing, dosage, and sequence, combined with the relatively small sample size, limited the possibility of meaningful subgroup analyses.

## 5. Conclusions

In conclusion, prolonged COVID-19 pneumonia in patients with hematologic malignancies is associated with the use of antibody-based drugs targeting the B-cell lineage and results in longer hospital stays and delays in treatments for the underlying malignancy. Prolonged COVID-19 pneumonia may persist and show radiologic worsening even after 90 days from the initial COVID-19 diagnosis in patients with hematologic malignancies.

## Figures and Tables

**Figure 1 jcm-14-02701-f001:**
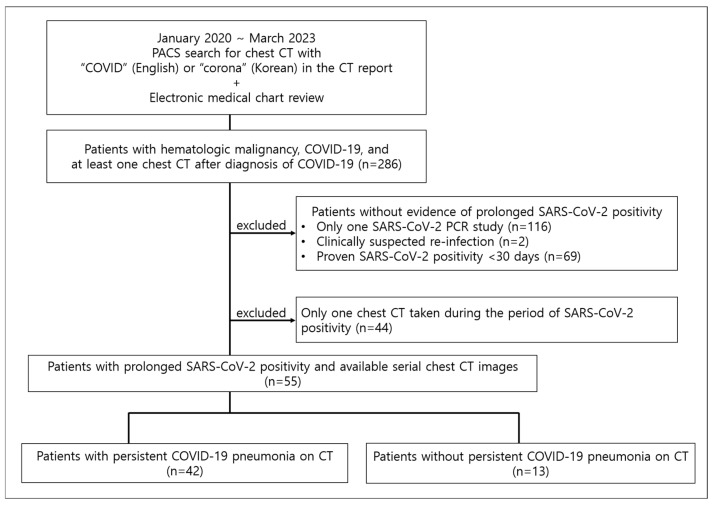
Study population. PACS = picture archiving and communication system.

**Figure 2 jcm-14-02701-f002:**
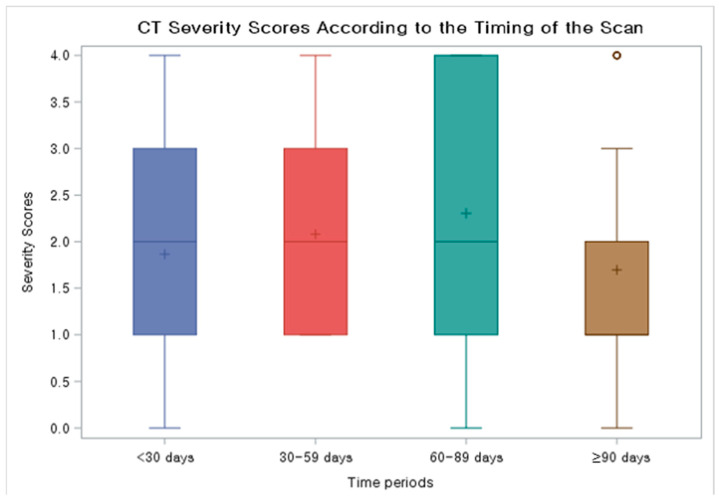
Box and whisker plot showing CT severity scores according to the timing of the scan. Median severity scores for CTs taken during < 30 days, 30–59 days, 60–89 days, and ≥90 days from initial diagnosis were 2.0 (IQR 1.0–3.0), 2.0 (IQR 1.0–3.0), 2.0 (IQR 1.0–3.5), and 1.0 (1.0–2.0), respectively.

**Figure 3 jcm-14-02701-f003:**
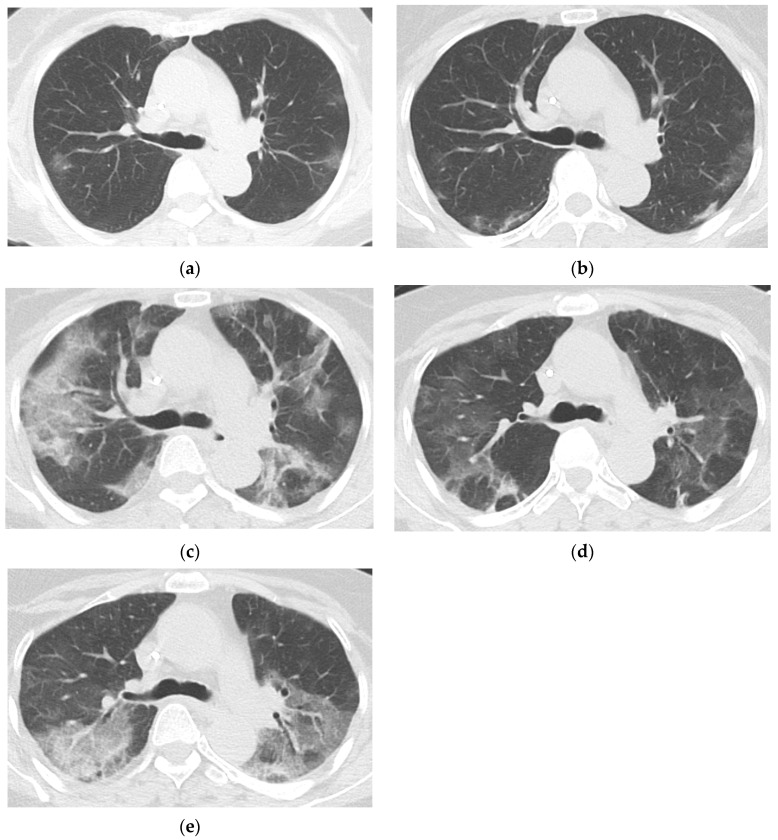
Prolonged COVID-19 pneumonia in a 65-year-old female with follicular lymphoma who had last received rituximab 117 days before COVID-19 diagnosis. Chest CTs of the patient taken 11 days (**a**), 20 days (**b**), 32 days (**c**), 46 days (**d**), and 61 days (**e**) after COVID-19 diagnosis show persistent COVID-19 pneumonia in the form of patchy peripheral and peribronchovascular ground glass opacities (GGOs) migrating in the upper lungs. The patient underwent SARS-CoV-2 PCR testing at 11, 21, 32, 36, 41, 48, 56, 63, and 69 days after COVID-19 diagnosis; all results were positive, with a median cycle threshold value of 12.71. She had last received rituximab, an anti-CD20 agent, 117 days before COVID-19 diagnosis.

**Figure 4 jcm-14-02701-f004:**
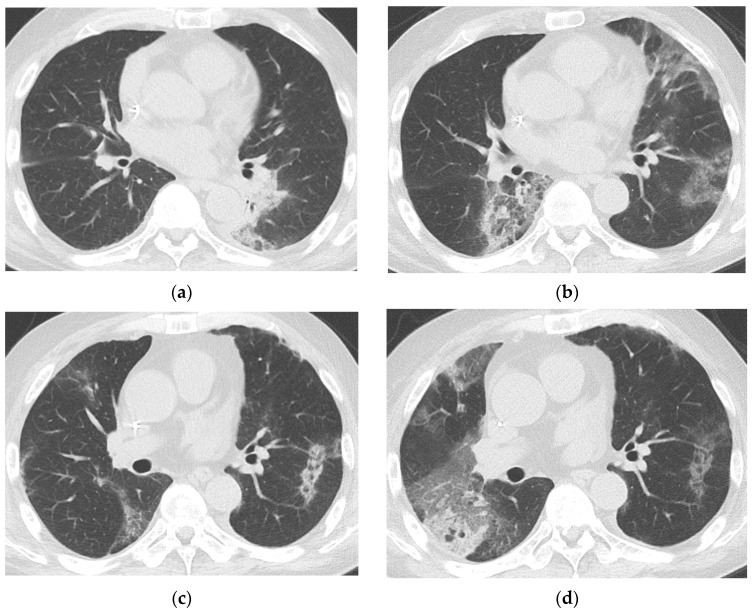
Prolonged COVID-19 pneumonia in a 64-year-old male with multiple myeloma who had last received teclistamab one day before COVID-19 diagnosis. Chest CTs of a 64-year-old male with multiple myeloma 15 days (**a**), 31 days (**b**), 44 days (**c**), and 52 days (**d**), respectively, after COVID-19 diagnosis show persistent COVID-19 pneumonia. Patchy peripheral and peribronchovascular consolidation and GGO in the left lower lobe on day 15 (**a**) completely resolved and new patchy peripheral GGOs appear in other areas on day 31 (**b**), consistent with migration. Most of these lesions decreased in extent on day 44 (**c**), but a new patchy GGO us seen in the right middle lobe. On day 52 (**d**), the extent of GGOs in the right lung had increased, while the lesion in the left lower lobe became fainter. The patient underwent SARS-CoV-2 PCR testing at 6, 13, 20, 27, 31, 38, 45, 52, 59, and 64 days after COVID-19 diagnosis; all results were positive, with a median cycle threshold value of 17.15. He had last received teclistamab, an antibody-based drug against B-cell maturation antigen, one day prior to COVID-19 diagnosis. The patient died of COVID-19 on day 67.

**Table 1 jcm-14-02701-t001:** Baseline characteristics of study patients.

Clinical Characteristics	All Patients (N = 55)	COVID-19 Pneumonia (N = 42)	No COVID-19 Pneumonia (N = 13)	*p* Value
Age, years	61 (51–65)	61 (51–65)	58 (52–63)	0.317
Sex, female: male	19 (34.5%):36 (65.5%)	15 (35.7%):27 (64.3%)	4 (30.8%):9 (69.2%)	>0.999
Hematologic malignancy				0.011
Leukemia	14 (25.5%)	7 (16.7%)	7 (53.8%)	
Lymphoma	30 (54.5%)	25 (59.5%)	5 (38.5%)	
Multiple myeloma	9 (16.4%)	9 (21.4%)	0 (0.0%)	
Myelodysplastic syndrome	2 (3.6%)	1 (2.4%)	1 (7.7%)	
Treatments for hematologic malignancy				
B-cell-directed antibody-based therapies ^b^ within 1 year	36 (65.5%)	31 (73.8%)	5 (38.5%)	0.042
anti-CD20 agents ^a^ within 1 year	28 (50.9%)	24 (57.1%)	4 (30.8%)	0.097
hematopoietic stem cell transplant	15 (27.3%)	10 (23.8%)	5 (38.5%)	0.310
Vaccination against SARS-CoV-2				0.910
Yes	23 (41.8%)	17 (40.5%)	6 (46.2%)	
No	8 (14.5%)	6 (14.3%)	2 (15.4%)	
Unknown	24 (43.6%)	19 (45.2%)	5 (38.5%)	
Duration of SARS-CoV-2 PCR positivity, days	69 (55–100.5)	72 (56–108)	65 (54–70)	0.106
Number of SARS-CoV-2 PCR tests performed	9 (5–11.5)	10 (6–13)	6 (4–9)	0.055
Time interval between SARS-CoV-2 PCR tests, days	7 (6–13)	7 (5–13)	7 (7–11)	0.425
Median cycle threshold value during the disease course				0.179
<20	30 (54.5%)	25 (59.5%)	5 (38.5%)	
20–30	23 (41.8%)	17 (40.5%)	6 (46.2%)	
>30	2 (3.6%)	0 (0.0%)	2 (15.4%)	
Presumed SARS-CoV-2 subtype				0.425
Original strain	3 (5.5%)	3 (7.1%)	0 (0.0%)	
Delta variant	2 (3.6%)	2 (4.8%)	0 (0.0%)	
Omicron variant	50 (90.9%)	37 (88.1%)	13 (100.0%)	
Delay in treatment for hematologic malignancy				0.215
Yes	34 (61.8%)	27 (64.3%)	6 (46.2%)	
No	8 (14.5%)	4 (9.5%)	4 (30.8%)	
Further treatment not planned	13 (23.6%)	11 (26.2%)	3 (23.1%)	
Duration of treatment delay for hematologic malignancy, days	43 (13–426)	49 (39–104)	29.5 (21–41)	0.027
Symptoms during persistent SARS-CoV-2 infection				0.080
Yes ^c^	50 (87.3%)	40 (95.2%)	10 (76.9%)	
No	5 (12.7%)	2 (4.8%)	3 (23.1%)	
Total number of hospital admissions for COVID-19	1 (1–1)	1 (1–1)	1 (0–1)	0.005
Total length of hospital stay for COVID-19	12 (7–33)	20 (8–41)	6 (0–12)	0.002
Medical treatment for COVID-19				
Systemic corticosteroids	36 (65.5%)	33 (78.6%)	3 (23.1%)	<0.001
Antiviral therapies	38 (69.1%)	28 (66.7%)	10 (76.9%)	0.733
Remdesivir	32 (58.2%)	27 (64.3%)	5 (38.5%)	
Nirmatrelvir/ritonavir or Molnupiravir	13 (23.7%)	7 (16.7%)	6 (46.2%)	
ICU admission and ventilator care	6 (10.9%)	6 (14.3%)	0 (0.0%)	0.321
Death				
COVID-19-specific mortality	14 (25.5%)	13 (31.0%)	1 (7.7%)	0.147
30-day all-cause mortality	21 (38.2%)	16 (38.1%)	5 (38.5%)	>0.999

Data are median (range) or number of patients (percentage). ICU = intensive care unit, PCR = polymerase chain reaction. ^a^ obinutuzumab, odronextamab, or rituximab; ^b^ belantamab mafodotin, daratumumab, elranatamab, inotuzumab ozagomicin, linvoseltamab, obinutuzumab, odronextamab, rituximab, or teclistamab; ^c^ persistent or relapsing fever, cough, dyspnea, sputum, sore throat, etc.

**Table 2 jcm-14-02701-t002:** Antibody-based drugs targeting B-cell lineage used by study patients.

Antibody-Based Drug	Type	Target	Indication	Number of Patients	Number of Patients with Prolonged COVID-19 Pneumonia
Rituximab	Monoclonal antibody	CD20	B-NHL	24	20
Odronextabmab	Bispecific antibody	CD20/CD3	B-NHL, DLBCL	3	3
Daratamumab ^a^	Monoclonal antibody	CD38	MM	3	3
Elranatamab	Bispecific antibody	BCMA/CD3	MM	2	2
Inotuzumab ozogamicin	Antibody–drug conjugate	CD22	B-NHL, B-ALL	2	1
Belantamab mafodotin ^b^	Antibody–drug conjugate	BCMA	MM	1	1
Linvoseltamab ^b^	Bispecific antibody	BCMA/CD3	MM	1	1
Obinutuzumab	Monoclonal antibody	CD20	DLBCL, MCL, FL, CLL	1	1
Teclistamab ^c^	Bispecific antibody	BCMA/CD3	MM	2	2
Total				36	31

B-ALL = B-cell acute lymphoblastic leukemia, BCMA = B-cell maturation antigen, B-NHL = B-cell non-Hodgkin lymphoma, CLL = chronic lymphocytic leukemia, DLBCL = diffuse large B-cell lymphoma, FL = follicular lymphoma, MCL = mantle cell lymphoma, MM = multiple myeloma. ^a^ Each of three patients had additionally received belantamab mafodotin, linvoseltamab, and teclistamab, respectively, after receiving daratamumab. ^b^ The patient had also received daratamumab prior to receiving this drug. ^c^ One patient had also received daratamumab prior to receiving this drug.

**Table 3 jcm-14-02701-t003:** Univariable and multivariable logistic regression analyses for development of prolonged COVID-19 pneumonia.

Variable	Univariable Analysis		Multivariable Analysis	
	OR (95% CI)	*p* Value	Adjusted OR (95% CI)	*p* Value
Age	1.03 (0.98–1.08)	0.250		
Type of hematologic malignancy (vs. leukemia)				
Lymphoma	5.0 (1.2–20.71)	0.966		
Multiple myeloma	>999.99 (<0.001–>999.99)	0.947		
Myelodysplastic syndrome	1.0 (0.05–19.4)	0.938		
Use of antibody-based drugs targeting B-cell lineage within 1 year	4.51 (1.21–16.75)	0.025	4.34 (1.06–17.81)	0.041
Use of Anti-CD20 agent within 1 year	3.00 (0.80–11.31)	0.105		
Receiving hematopoietic stem cell transplant	0.50 (0.13–1.88)	0.305		
Duration of SARS-CoV-2 positivity	1.02 (1.00–1.04)	0.119		
Presence of symptoms	6.00 (0.88–40.87)	0.067	6.10 (0.74–50.17)	0.093
Median Ct value	0.88 (0.76–1.02)	0.092	0.92 (0.79–1.08)	0.305

Ct = cycle threshold, CI = confidence interval, OR = odds ratio.

**Table 4 jcm-14-02701-t004:** Baseline and follow-up chest CT findings of 42 patients with prolonged COVID-19 pneumonia.

Time of CT Exam (Days from the 1st Positive PCR)	<30 Days (N = 53)	30–59 Days (N = 50)	60–89 Days (N = 23)	≥90 Days (N = 33)
Median number of days from the 1st positive PCR	17	41.5	72	124
RSNA CT categorization of COVID-19 pneumonia				
Typical	37 (69.8%)	39 (78.0%)	16 (69.6%)	18 (54.5%)
Indeterminate	8 (15.1%)	9 (18.0%)	5 (21.7%)	7 (21.2%)
Atypical	6 (11.3%)	2 (4.0%)	2 (8.7%)	4 (12.1%)
Negative	2 (3.8%)	0 (0.0%)	0 (0.0%)	4 (12.1%)
Pattern				
GGO	23 (43.4%)	18 (36.0%)	11 (47.8%)	16 (48.5%)
Consolidation	4 (7.5%)	2 (4.0%)	3 (13.0%)	5 (15.2%)
Both	14 (26.4%)	27 (56.0%)	9 (39.1%)	8 (24.2%)
Distribution				
Peripheral	8 (15.1%)	7 (14.0%)	1 (4.3%)	2 (6.1%)
Peribronchovascular	9 (17.0%)	4 (8.0%)	4 (17.4%)	3 (9.1%)
Both	34 (64.2%)	37 (74.0%)	18 (78.3%)	24 (72.7%)
Severity				
0% involvement	2 (3.8%)	0 (0.0%)	0 (0.0%)	4 (12.1%)
1–25% involvement	24 (45.3%)	16 (32.0%)	8 (34.8%)	15 (45.5%)
26–50% involvement	12 (22.6%)	16 (32.0%)	5 (21.7%)	7 (21.2%)
51–75% involvement	12 (22.6%)	15 (30.0%)	4 (17.4%)	1 (3.0%)
76–100% involvement	3 (5.7%)	3 (6.0%)	6 (26.1%)	6 (18.2%)
Change in Extent				
Decreased	5 (9.4%)	12 (24.0%)	7 (30.4%)	8 (24.2%)
Stable	2 (3.8%)	9 (18.0%)	2 (8.7%)	14 (42.4%)
Increased	10 (18.9%)	25 (50.0%)	12 (52.2%)	11 (33.3%)
Unassessable	36 (67.9%)	4 (8.0%)	2 (8.7%)	0 (0.0%)
Migration				
Yes	5 (9.4%)	8 (16.0%)	5 (21.7%)	5 (15.2%)
No	12 (22.6%)	38 (76.0%)	16 (69.6%)	28 (84.8%)
Unassessable	36 (67.9%)	4 (8.0%)	2 (8.7%)	0 (0.0%)

## Data Availability

The datasets generated or analyzed during the study are available from the corresponding author upon reasonable request.

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
