# Peer review of "Prolonged COVID-19 Pneumonia in Patients with Hematologic Malignancies: Clinical Significance and Serial CT Findings"

_jcm, 2025, doi:10.3390/jcm14082701_

Round 1

Reviewer 1 Report

Comments and Suggestions for Authors

The manuscript titled "Prolonged COVID-19 Pneumonia in Patients with Hematologic Malignancies: Clinical Significance and Serial CT findings," authored by Dae Hee Han, Raeseok Lee, Gi June Min, Jongmin Lee, Yejin Sohn, Eun Jeong Min, Jinyoung Lee, Jung Im Jung, and Kyongmin Sarah Beck, addresses an important clinical knowledge gap regarding prolonged COVID-19 pneumonia in hematologic malignancy patients treated with B-cell-directed therapies. However, the manuscript requires modifications depending on the following comments:

Comments:

  1. The manuscript uses the term "migration" of pneumonia. To make the study more reliable, the manuscript should give a clearer explanation and set of criteria for measuring the "migration" of pneumonia than what is shown in the figures.
  2. The manuscript may include alternative methods, such as time-to-event analysis, and explain the choice of logistic regression.
  3. The manuscript could include a large sample size to conclude the outcome. A single-center retrospective design (n=55, minor sample size) is restricted; this could mislead the outcome. However, it could also discuss whether these findings align with those of other centers.
  4. The manuscript's significant observation is that the patient's radiologic opacities and treatments appeared unrelated (lines 172-173) and required more rigorous statistical analysis despite the challenges posed by treatment variability.
  5. The manuscript uses a CT severity scoring system (0-4 scale) and may benefit from adopting a more detailed scoring system or clarifying the rationale for selecting a CT severity scale.
  6. The manuscript could include a relationship between CT findings and clinical outcomes.
  7. The manuscript, highlighting a differing finding on GGO patterns (mixed and consolidation GGO, line 246), needs a more thorough exploration of potential explanations beyond simply differences in study design.
  8. The manuscript could include specific guidance for managing these patients, particularly regarding when to resume cancer treatment.
  9. The manuscript's CT images could benefit by adding scale bars and highlighting the specific areas of migration and change by adding arrows in Figures 2 and 3.
  10. Formatting Table 1 could improve it to prevent inconsistencies.
  11. The manuscript could include precision measures and confidence intervals in statistical results.
  12. The manuscript could be defined upon the first use of several acronyms, such as BCMA and GGO, in the manuscript.
  13. The manuscript requires additional context for the term "organizing pneumonia" (Line 278).
  14. Table 4 can be formatted with more precise column lines and column headings.
Comments on the Quality of English Language
  1. Please complete the sentence of line 321.
  2. Please correct -." uthors should discuss the results" (line 400).
  3. The manuscript needs to be rechecked for grammar.

Author Response

Modified parts are indicated in the revised manuscript with memos.

  1. The manuscript uses the term "migration" of pneumonia. To make the study more reliable, the manuscript should give a clearer explanation and set of criteria for measuring the "migration" of pneumonia than what is shown in the figures.
  • Thank you for this valuable comment. We acknowledge the need for a clearer definition and criteria for "migration" of pneumonia. However, “migration” is a variable with a binary outcome, so it’d be difficult to measure. In the revised manuscript, we have more clearly defined "migration" as the emergence of new airspace opacities in different lung regions accompanied by the resolution of previous lesions seen on previous CT. A specific timeframe could not be defined, because CTs of patients were taken arbitrarily, with the need and timing of the follow-up exam decided by the clinicians (Reviewer 1, Comment 1).

2. The manuscript may include alternative methods, such as time-to-event analysis, and explain the choice of logistic regression.

  • We appreciate the suggestion to consider alternative statistical methods. While we acknowledge that time-to-event analysis could provide additional insights, our primary objective was to identify factors associated with the development of prolonged COVID-19 pneumonia. Logistic regression was chosen due to its suitability for analyzing binary outcomes in small sample sizes.

3. The manuscript could include a large sample size to conclude the outcome. A single-center retrospective design (n=55, minor sample size) is restricted; this could mislead the outcome. However, it could also discuss whether these findings align with those of other centers.

  • We sincerely appreciate the reviewer’s insightful comment. We acknowledge the limitations inherent in a single-center retrospective study. However, prolonged COVID-19 pneumonia represents a clinically significant yet relatively rare condition, primarily affecting a highly selective group of immunocompromised patients. As a result, most available literature consists of case reports, and the inclusion of 55 patients in our study represents a substantial cohort given the rarity of this condition. While a larger, multicenter study would be ideal, our study provides valuable initial data on this underexplored topic. We have explicitly acknowledged this limitation in the revised manuscript and have emphasized the need for future studies with larger cohorts in the limitations section of the discussion (Reviewer 1, Comment 3).

4. The manuscript's significant observation is that the patient's radiologic opacities and treatments appeared unrelated (lines 172-173) and required more rigorous statistical analysis despite the challenges posed by treatment variability.

  • We truly appreciate this insightful comment. We agree that further statistical analysis would be valuable. However, the substantial variability in the timing, dosage, and sequence of individual treatments presents significant challenges for conducting a meaningful subgroup analysis. Additionally, the relatively small sample size further limits the feasibility of such an analysis. We have explicitly addressed this limitation in the revised manuscript's discussion section (Reviewer 1, Comment 4).

5. The manuscript uses a CT severity scoring system (0-4 scale) and may benefit from adopting a more detailed scoring system or clarifying the rationale for selecting a CT severity scale.

  • We sincerely appreciate the reviewer’s comment. The CT severity scoring system (0–4 scale) was selected for its simplicity and reproducibility. Additionally, a previous systematic review on prolonged SARS-CoV-2 infection (Korean J Radiol 2024, 25, 473-480) utilized the same scoring system and found that severity scores remained persistently intermediate throughout the disease course. Given these findings, we anticipated similar results in our study and therefore did not consider a more detailed but complex scoring system necessary. We have now included a justification for this selection in the Materials and Methods section of the revised manuscript (Reviewer 1, Comment 5).

6. The manuscript could include a relationship between CT findings and clinical outcomes.

  • We appreciate this important suggestion. We did try to address the relationship between CT findings and clinical outcomes by analyzing the relationship between number of inpatient admissions, length of inpatient, delays in treatments for hematologic malignancies, COVID-19 specific mortality, and 30-day all-cause mortality in days, and the presence of CT findings compatible with prolonged COVID-19 pneumonia. Our results show that patients showing CT findings of prolonged COVID-19 pneumonia are associated with more frequent inpatient admissions, longer inpatient days for COVID-19, and longer delays in treatments for hematologic malignancies; COVID-19 specific mortality and 30-day all-cause mortality were not statistically different between those showing prolonged COVID-19 pneumonia on CT and those without signs of prolonged COVID-19 pneumonia on CT (Reviewer 1, Comment 6).

7. The manuscript, highlighting a differing finding on GGO patterns (mixed and consolidation GGO, line 246), needs a more thorough exploration of potential explanations beyond simply differences in study design.

  • We acknowledge the need for a more in-depth discussion regarding the differences in CT patterns observed in our study. In addition to study design differences, we believe that the much longer follow-up period (median number of days after initial PCR positivity for the longest follow-up being 89 days vs. 124 days) in our study could have also affected the CT findings. We have added this in the discussion of the revised manuscript (Reviewer 1, Comment 7).

8. The manuscript could include specific guidance for managing these patients, particularly regarding when to resume cancer treatment.

  • We appreciate this insightful comment. We agree that determining the appropriate timing for resuming cancer treatment in patients with persistent COVID-19 is a critical clinical issue. However, the primary objective of our study was to characterize the clinical and radiologic features of prolonged COVID-19 rather than to establish treatment resumption guidelines. While our data do not allow us to provide specific recommendations, we recognize the significance of this decision and have briefly addressed it in the Discussion section of the revised manuscript as follows: "Resuming cancer treatment in patients with persistent COVID-19 requires a comprehensive evaluation of multiple factors, including SARS-CoV-2 PCR results, clinical and radiologic improvement, and the intensity and urgency of planned cancer therapy." (Reviewer 1, Comment 8).

9. The manuscript's CT images could benefit by adding scale bars and highlighting the specific areas of migration and change by adding arrows in Figures 2 and 3.

  • We appreciate this valuable comment. Scale bars are not typically required in CT images, as they inherently provide anatomical context and proportionality. Additionally, we opted not to include arrows to avoid cluttering the images, as the airspace opacities and their changes over time are clearly visible. We believe this approach maintains the clarity and readability of the figures.

10. Formatting Table 1 could improve it to prevent inconsistencies.

  • Thank you for this valuable comment. Table 1 was formatted to dispose of any inconsistencies (Reviewer 1, Comment 10).

11. The manuscript could include precision measures and confidence intervals in statistical results.

  • We appreciate this suggestion. To enhance statistical rigor, we have included precision measures and confidence intervals in our statistical results where applicable, in lines 215-218 and Table 3, of the revised manuscript (Reviewer 1, Comment 11).

12. The manuscript could be defined upon the first use of several acronyms, such as BCMA and GGO, in the manuscript.

  • Thank you for this valuable comment. We have defined the acronyms at their first use in the revised manuscript (Reviewer 1, Comment 12).

13. The manuscript requires additional context for the term "organizing pneumonia" (Line 278).

  • We acknowledge the need for more context regarding the term "organizing pneumonia." In the revised manuscript, we have added an explanation on why some may argue these migrating opacities to be findings of organizing pneumonia (Reviewer 1, Comment 13).

14. Table 4 can be formatted with more precise column lines and column headings.

  • We appreciate this comment. Table 4 has been reformatted with clearer column lines and headings to enhance readability and ensure precise data presentation (Reviewer 1, Comment 14).

15. Please complete the sentence of line 321.

  • Thank you for this valuable comment. The sentence was an irrelevant sentence included as a mistake. It was deleted in the revised manuscript (Reviewer 1, Comment 15).

16. Please correct -." uthors should discuss the results" (line 400).

  • Thank you for this valuable comment. The sentences in lines 400-403 are also irrelevant sentences included as a mistake. They were deleted in the revised manuscript (Reviewer 1, Comment 16).

17. The manuscript needs to be rechecked for grammar.

  • Thank you for this valuable comment. The manuscript was rechecked for grammar, and appropriate grammatical modifications were made in the revised manuscript.

We thank the reviewers for their constructive feedback, which has significantly improved the quality and clarity of our manuscript. We hope that the revised version meets the expectations of the reviewers and the journal.

Reviewer 2 Report

Comments and Suggestions for Authors

Thank you for the opportunity to review this paper. The authors present the results of persistent COVID in patients with hematological malignancies. This is a very important topic, that is still highly relevant today, even though the COVID strains became weaker. There are some very interesting results, although I think the article has significant limitations and issues.

  1. Abstract – Please add the study design (retrospective). In addition, what do you mean by "showed COVID-19 pneumonia on CT"? This is obviously not a radiologic sign, compared to GGOs/opacities etc. Also what do you mean by "median CT duration"? is it time from first to second? All comparisons are made to which control group – please state it.
  2. Methods –

 - Please delete line 321

 - This section should begin with something like – this is a retrospective study, conducted at XXXX, including XXX. Then continue with the text.

 - The fact that you excluded 42 patients due to only one CT is a major limitation. This population has persistent COVID and it is almost the size of your cohort size. This might have altered the results, as those who had more than one CT might have done it due to more severe disease or complications, hence major bias in your results.

 - Another limitation which I think should be addressed in the results is the difference in COVID strains over the years. This is a major issue, as the omicron results in much milder disease. Please specify how many patients per strain (based on the time they were infected or if you know from cultures).

- Why were patients with COVID-19 pneumonia were hospitalized? Why did they perform Ct scan (and why more than 1?)?

  1. Results:

 - How can 38% die in 30 days and still has persistent COVID and follow-up of over 30 days?

 - Were there any other reasons for immunosuppression in any of the patients?

 -   Please explain how you divided patients to the two groups – only if a patient had signs of COVID on all his CT scans he was in the positive group?

 - You should not use a multivariate analysis when only 13 patients are in one of the groups. If you want- include ONLY the factors significant from the univariate analysis – 3-4 max given the sample size.

  1. Discussion:

 - An interesting issue is the very prolong pneumonia in most patients. This must be taken into consideration by physicians when meeting such patient with fever, as it can often not be taken into account if a lot of time has passed, resulting in missed diagnosis. Another very important issue is that even though 9 patients had COVID with pulmonary findings – in the end they had fungal pneumonia. This highlights the diagnostic challenge in COVID-19 patients. A large paper found 30% of missed or delayed diagnosis in such cases (DOI: 10.1002/jhm.13063). I suggest the authors to use this example to support their writings on this issue.  

Author Response

Modified parts are indicated in the revised manuscript with memos.

Thank you for the opportunity to review this paper. The authors present the results of persistent COVID in patients with hematological malignancies. This is a very important topic, that is still highly relevant today, even though the COVID strains became weaker. There are some very interesting results, although I think the article has significant limitations and issues.

  1. Abstract – Please add the study design (retrospective). In addition, what do you mean by "showed COVID-19 pneumonia on CT"? This is obviously not a radiologic sign, compared to GGOs/opacities etc. Also what do you mean by "median CT duration"? is it time from first to second? All comparisons are made to which control group – please state it.
  • Thank you for these valuable comments. We added the study design (retrospectively) in the abstract of the revised manuscript. The sentence was modified to “76.4% of patients had presence of COVID-19 pneumonia on CT, which was prolonged with 35.5 days as a median CT duration of pneumonia, and they experienced more (p=0.005) and longer (p=0.002) hospital stays and longer de-lays in treatment for underlying malignancy (p=0.03), compared to those without evidence of COVID-19 pneumonia on CT.” in the revised manuscript in order to clarify our points. We have stated in the 4.2. Image Analysis of the Materials and Methods that “Total duration of pneumonia was calculated by counting the days between a patient’s first and last CTs with evidence of COVID-19 pneumonia.” (Reviewer 2, Comment 1).

Methods –

 - Please delete line 321

  • Thank you for checking this. The line was deleted (Reviewer 2, Comment 2).

 - This section should begin with something like – this is a retrospective study, conducted at XXXX, including XXX. Then continue with the text.

  • We appreciated this comment. “This retrospective study was approved by the Institutional Review Board of Seoul St. Mary’s Hospital (approval number: KC23RISI0347). The requirement for written informed consent was waived.” has been added to the Materials and Methods section of the revised manuscript (Reviewer 2, Comment 3).

 - The fact that you excluded 42 patients due to only one CT is a major limitation. This population has persistent COVID and it is almost the size of your cohort size. This might have altered the results, as those who had more than one CT might have done it due to more severe disease or complications, hence major bias in your results.

  • Thank you for this valuable comment. We also agree that this is a major limitation; but since our study aimed to analyze serial changes in CT findings, the inclusion of patients with only a single scan was not feasible. We have added a thorough explanation in the limitation in the discussion section of the revised manuscript. (Reviewer 2, comment 4)

 - Another limitation which I think should be addressed in the results is the difference in COVID strains over the years. This is a major issue, as the omicron results in much milder disease. Please specify how many patients per strain (based on the time they were infected or if you know from cultures).

  • We appreciate this insightful comment. Heeding your advice, we added the presumed SARS-CoV-2 subtypes in Table 1. We used presumed SARS-CoV-2 subtypes based on the most dominant variant in South Korea at the date of initial PCR positivity, the because none of the study patients had undergone virus cultures. These were added in materials and methods and table 1 of the revised manuscript (Reviewer 2, Comment 5).

- Why were patients with COVID-19 pneumonia were hospitalized? Why did they perform Ct scan (and why more than 1?)?

  • Thank you for this valuable comment. The patients were mostly hospitalized for their ongoing symptoms and the need for intravenous remdesivir therapy. The patients performed more than one CT scans at the discretion of the clinicians, to follow up on the existing lesions or to evaluate for their ongoing symptoms

Results:

 - How can 38% die in 30 days and still has persistent COVID and follow-up of over 30 days?

  • We defined 30-day all-cause mortality as “any death within 30 days from the last day of COVID-19 PCR positivity,” as explained in the materials and methods (lines 158-159). All included patients had last COVID-19 PCR positivity more than 30 days from the initial COVID-19 PCR positivity, and those who had died within 30 days from that last COVID-19 PCR positivity were regarded as 30-day all-cause mortality (Reviewer 2, Comment 7).

 - Were there any other reasons for immunosuppression in any of the patients?

  • Thank you for this valuable comment. No there weren’t any other reasons for immunosuppression in these patients other than hematologic malignancies.

 -   Please explain how you divided patients to the two groups – only if a patient had signs of COVID on all his CT scans he was in the positive group?

  • We appreciate this valuable comment. If the patients had presence of COVID-19 pneumonia on any of the CTs, they were grouped as “COVID-19 pneumonia group, ” and if the patients didn’t show any evidence of COVID-19 pneumonia on any of the CTs, they were grouped as “No COVID-19 pneumonia group.” To clarify this point, sentences were modified in the revised manuscript (Reviewer 2, comment 9).

 - You should not use a multivariate analysis when only 13 patients are in one of the groups. If you want- include ONLY the factors significant from the univariate analysis – 3-4 max given the sample size.

  • We appreciate this valuable comment. We were also aware of the limitations posed by the small sample size (n=13) of one of the groups, so we only included three factors with a p value of 0.10 or less into the multivariate analysis. We have clarified this in the statistical analysis section in the revised manuscript (Reviewer 2, Comment 10).
  1. Discussion:

 - An interesting issue is the very prolong pneumonia in most patients. This must be taken into consideration by physicians when meeting such patient with fever, as it can often not be taken into account if a lot of time has passed, resulting in missed diagnosis. Another very important issue is that even though 9 patients had COVID with pulmonary findings – in the end they had fungal pneumonia. This highlights the diagnostic challenge in COVID-19 patients. A large paper found 30% of missed or delayed diagnosis in such cases (DOI: 10.1002/jhm.13063). I suggest the authors to use this example to support their writings on this issue.

  • Thank you so much for this insightful comment. As you suggested, we have added discussion on how six of 13 patients without COVID-19 pneumonia were diagnosed with fungal pneumonia, and given that a previous study reported a secondary diagnosis in 17% of COVID-19 patients, maintaining a high index of suspicion for coexisting conditions is crucial, as is for prolonged SARS-CoV-2 infection in the discussion section of the revised manuscript (Reviewer 2, Comment 11).

We thank the reviewers for their constructive feedback, which has significantly improved the quality and clarity of our manuscript. We hope that the revised version meets the expectations of the reviewers and the journal.

Round 2

Reviewer 2 Report

Comments and Suggestions for Authors

Thank you for the opportunity to review this paper once again. The authors have thoroughly addressed all my comments and the article has significantly improved. I do not have any additional significant revision that is required. I do think some additional English corrections would improve the manuscript further.  

Comments on the Quality of English Language

See above

Author Response

Thank you for the opportunity to review this paper once again. The authors have thoroughly addressed all my comments and the article has significantly improved. I do not have any additional significant revision that is required. I do think some additional English corrections would improve the manuscript further.  

--> Thank you for reviewing our manuscript. As you suggested, we went over the manuscript for English corrections to improve the clarity for the readers. We hope that the revised version meets the expectations of the reviewers and the journal.